# Chemoenzymatic total synthesis of sorbicillactone A
Jonas I. Müller[1] & Tobias A. M. Gulder [1,2] ✉

The sorbicillinoid family is a large class of natural products known for their structural variety and strong, diverse biological activities. A special member of this family, sorbicillactone A, the first nitrogen-containing sorbicillinoid, exhibits potent anti-leukemic and anti-HIV activities and possesses a unique structure formed from sorbicillinol, alanine, and fumaric acid building blocks. To facilitate in-depth biological and structure-activity relationship studies of this promising natural product, we developed a chemoenzymatic approach that provides access to sorbicillactone A and several analogs with excellent yields under precise stereochemical control. The key steps of the highly convergent, stereoselective, and short route are the enantioselective oxidative dearomatization of sorbillin to sorbicillinol catalyzed by the enzyme SorbC and the subsequent Michael addition of a fumarylazlactone building block. Additionally, our synthetic findings and bioinformatic analysis suggest that sorbicillactone A is biosynthetically formed analogously.

The sorbicillinoids are a very large family of natural products with over a hundred isolated derivatives[1]. Due to their often highly complex three-dimensional structures and wide range of biological activities, sorbicillinoids have attracted considerable attention from various fields of the scientific community. The biosynthesis of all sorbicillinoids involves the same biosynthetic key step—the oxidative dearomatization of sorbicillin (**1**) to sorbicillinol (**2**) catalyzed by the monooxygenase SorbC[2–5]. The structural variety of the natural product class originates from the versatile reactivity of sorbicillinol (**2**), which can act as a Michael acceptor and donor in conjugate additions or as a diene and dienophile in Diels–Alder reactions. While a major part of the family consists of different dimeric structures, such as sorbiquinol (**3**)[6] and trichodimerol (**4**)[7], there are also hybrid structures containing additional biogenic building blocks, as found in sorbicatechol (**5**)[8] or sorbifuranone A (**6**)[9] (Fig. 1a).

The first nitrogen-containing sorbicillinoid, sorbicillactone A (**7**), was isolated in 2003 by Bringmann et al. from a sponge-derived *Penicillium chrysogenum* strain[10]. Biological evaluation of **7** revealed not only a selective activity against L5178y murine leukemic lymphoblasts but also strong anti-HIV activity by inhibiting viral protein expression and protecting human T lymphocytes against the cytopathic effect of HIV-1, as well as an effect on $[Ca^{2+}]_i$ in primary neurons, indicating possible neuroprotective properties. Structurally, the compound consists of a sorbicillinol (**2**), an alanine, and a fumaric acid unit fused to form an unusual bicyclic lactone structure.

Feeding experiments with $^{13}C$-labeled alanine and fumaric acid showed that the C3-unit forming the lactone indeed originated from alanine. However, no labeling was observed at the fumaryl side chain, indicating that fumaric acid is not a direct biosynthetic precursor. It was assumed that the addition of an activated species of alanine to sorbicillinol (**2**) and subsequent coupling of a fumaryl-related C4 unit leads to the formation of **7**[11].

For a long time, synthetic efforts targeting sorbicillinoids mainly focused on the (stereoselective) synthesis of **2** since this crucial biosynthetic intermediate also offers the most convenient synthetic access to this natural product class. All these efforts either lacked stereoselectivity or were long, low-yielding sequences[12–15]. Our group established a concise, efficient, and stereoselective biocatalytic approach towards **2**. Starting from 2-methylresorcinol, **1** can readily be synthesized in three steps and enantioselectively converted to **2** using the recombinantly produced enzyme SorbC[16]. This approach offers access to various sorbicillinoids, including **3**, **4**, and **5**[17–20]. Due to its unusual structure, a potential synthesis of **7** must overcome challenges not typical for other sorbicillinoids, particularly the stereoselective installation of three contiguous stereogenic centers at C5, C6, and C9. It is important to note that the original producer of sorbicillactone A (**7**) also produces a structurally highly related analog, sorbicillactone B, which is saturated at C2'/C3'. This compound has almost identical chromatographic properties and is thus highly tedious to be separated from **7**, in particular at the preparative scale[21].

[1]Chair of Technical Biochemistry, Technical University of Dresden, Bergstraße 66, 01069 Dresden, Germany. [2]Helmholtz Institute for Pharmaceutical Research Saarland (HIPS), Department of Natural Product Biotechnology, Helmholtz Centre for Infection Research (HZI) and Department of Pharmacy at Saarland University, 66123 Saarbrücken, Germany. ✉e-mail: tobias.gulder@tu-dresden.de

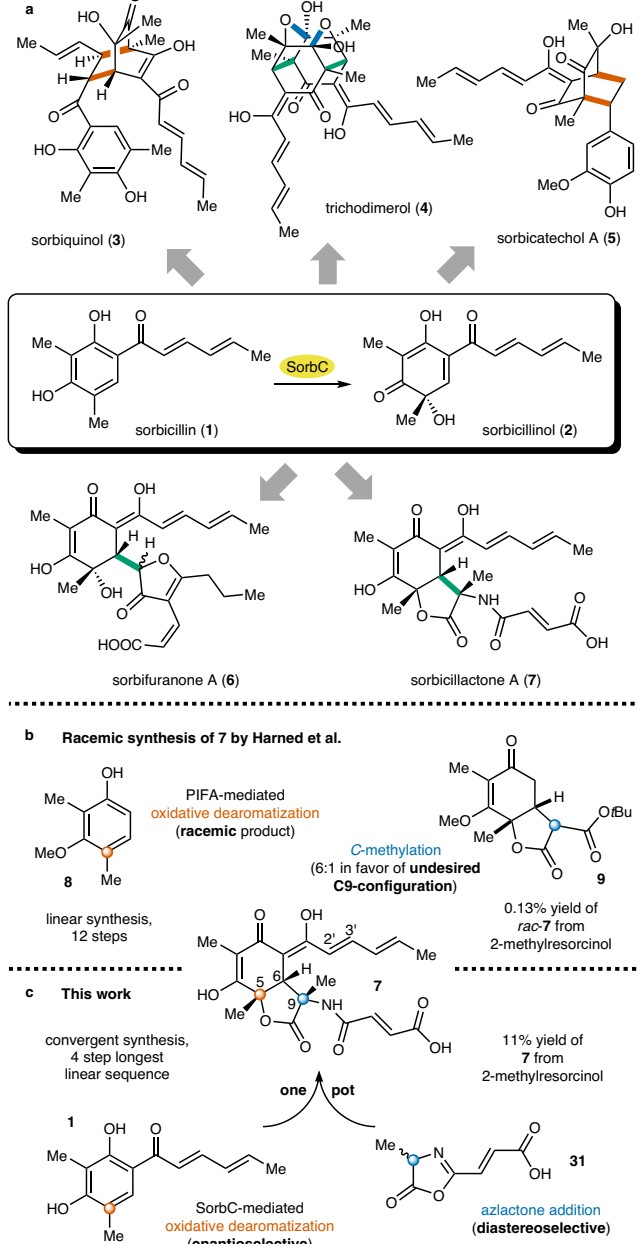

**Fig. 1 | The chemistry of sorbicillinoids. a** Selection of dimeric (**3,4**) and hybrid (**5–7**) sorbicillinoids formed by Michael additions (green bonds), Diels–Alder cycloadditions (red bonds), and ketalization (blue bonds) reactions after enantioselective oxidative dearomazation of **1** to **2** catalyzed by monooxygenase SorbC (labeled in yellow). **b** Key steps of the first total synthesis by Harned et al. yielding *rac*-**7**[22]. **c** Key synthetic steps of the current work delivering sorbicillactone A (**7**).

This makes efficient synthetic access to **7** desperately needed for future biomedical studies and compound development. The first and to the best of our knowledge only total synthesis of **7** published in 2011 by Volp et al.[22] relied on the initial formation of the bicyclic lactone core structure bearing the three stereogenic centers, with the late-stage installation of the sorbyl side chain (see Fig. S3 for the entire synthetic route). Key steps of the synthesis (Fig. 1b) were the oxidative dearomatization of phenol **8** using PIFA, leading to the corresponding *racemic* C5 alcohol, and the *C*-methylation of lactone **9**, unfortunately proceeding with undesired diastereoselectivity mainly delivering the incorrect C9 epimer in a 6:1 ratio. While this pioneering work thus provided synthetic access to *rac*-**7** for the

first time, it also left significant room for improvement concerning stereoselectivity and overall efficiency (12 linear steps, 0.13% overall yield of *rac*-**7**). In this work, we developed a streamlined, convergent access to stereochemically pure **7** (4 steps, 11% yield of **7**). The installation of all stereogenic centers is achieved in a single reaction pot, enabled by an enzyme-mediated enantioselective dearomatization of sorbicillin (**1**) to sorbicillinol (**2**), combined with immediate diastereoselective interception of the latter using an azlactone nucleophile, e.g., **31** (Fig. 1c). This methodology provides stereoselective synthetic access to **7** and is also amenable to synthesizing different C9 alkyl sorbicillactone analogs.

## Results and discussion
### Initial synthetic efforts towards sorbicillactone A (7)

Given the synthetic challenge and the promising biological properties, we set out to develop a biocatalytic approach towards **7** employing enantioselective enzymatic oxidative dearomatization of **1** to **2** catalyzed by SorbC as the key step. In our retrosynthetic analysis (Fig. 2a), we considered work published by Nicolaou et al.[23], who demonstrated that treatment of *O*-acetylsorbicillinol with KHMDS leads to an intramolecular Michael addition resulting in a lactone analog of **7**, only devoid of the C9 methyl and aminofumaryl substituents. With this in mind, retrosynthetic removal of the *N*-fumaryl residue would deliver **10**, which, upon disconnection of the C6/C9-*C,C*-bond, results in ester **11**. We envisioned that intramolecular Michael addition, e.g., by applying methodology developed by Kazmaier et al.[24,25], should allow bond formation to **10**. Specifically, an *N*-protected amino acid residue attached to **11** should be converted into a zinc-chelated enolate by double deprotonation to install sufficient nucleophilicity for intramolecular cyclization. Suitable electron-withdrawing protective groups can be installed in **11** to facilitate enolate formation. By directly using the required amino acid as a building block, additional downstream interconversion chemistry can be reduced to a minimum. Furthermore, conjugate addition in **11** was expected to proceed with chirality transfer from C5 to at least position C6. Removal of the amino acid ester in **11** results in sorbicillinol (**2**), which in turn is enantioselectively accessible from **1** by enzymatic oxidative aromatization employing SorbC (cf. Figure 1). To explore suitable *N*-protective groups and efficient conditions for the intramolecular cyclization reaction, we first used a purely synthetic route to produce sufficient amounts of different analogs of **11**. The enzymatic oxidative dearomatization was hence initially replaced by a chemical transformation using PIFA (phenyliodine bis(trifluoroacetate)). This effective conversion had already been applied by Porco et al. in the synthesis of sorbiterrin A: activation of *O*-acetylsorbicillin with PIFA enabled an oxidative 1,2-acyl shift to give the desired dearomatized racemic *O*-acetylsorbicillinol[26]. Therefore, we set out to evaluate if this transformation can be utilized to rearrange more advanced acyl groups facilitating an assembly of **7**.

Starting from ʟ-alanine (**12**), we performed a TFA (trifluoroacetyl) protection of the amino group to obtain **13** in quantitative yield (Fig. 2b). Subsequently, we coupled **13** to sorbicillin (**1**) using Steglich esterification to yield **14** in 92%. Sorbicillin (**1**) was synthesized in 38% yield over three steps according to previous work starting from 2-methylresorcinol[16]. The PIFA-mediated oxidation indeed allowed for the dearomatization of **14** and the simultaneous 1,2-acyl shift of the attached alanine residue, yielding sorbicillinol derivative **15** in 29%. The latter was readily converted to the desired lactone **16a** (52%) by intramolecular Michael addition. We also isolated the undesired **16b** as the minor product in a 6% yield. Based on these encouraging results, we aimed to transfer this approach to stereochemically pure **2** (Fig. 2c). Therefore, sorbicillinol (**2**) was made enzymatically and intercepted in vitro with pentafluorophenol (pfp) activated *N*-TFA-protected ʟ-alanine. Surprisingly, this did not deliver ester intermediate **15** but led to direct conversion to stereochemically defined (5*S*, 6*R*, 9*S*)-**16a** in 18% yield. To our delight, we did not observe the formation of the corresponding

**Fig. 2 | Initial synthetic efforts towards sorbicillactone A. a** Initial retrosynthetic approach towards sorbicillactone A (7). **b** Unsuccessful synthesis of sorbicillactone derivative 17 devoid of the fumaryl side chain. **c** Chemoenzymatic synthesis of TFA-protected sorbicillactone derivative 16a using recombinant SorbC (labeled in yellow). **d** Synthesis of nitrosorbicillactones 21a/b.

C9 epimer (5S, 6R, 9R)-16b or any other stereoisomer, attesting to the possibility of utilizing such an approach for a highly stereoselective synthesis of 7. However, despite extensive experimentation, it was impossible to deprotect 16a/b to 17 under acidic, basic, or reductive conditions. All attempts either did not show any conversion or led to decomposition.

We next screened for alternative protection groups to replace *N*-TFA protection. Unfortunately, none of the tested esters 15 with different *N*-protection groups (e.g., SES, Boc, Cbz, or Pht) could be cyclized to the respective desired lactones 16. This is likely a result of the insufficient acidity at the amino acid alpha-carbon when using less electron-withdrawing protective groups compared to *N*-TFA. To circumvent the need for intermediate deprotection while facilitating alpha deprotonation, we next investigated the synthesis of the nitro derivative 21 (Fig. 2d). As 2-nitropropanoic acid is prone to decarboxylation, we envisioned the attachment of a building block to the sorbicillin core that would allow later installation of the desired nitro functionality. Therefore, we coupled 2-bromo propanoic acid (18) to sorbicillin (1) in 96% yield. The oxidation of 19 with PIFA yielded sorbicillinol derivative 20 in 32% yield. The nitration of 20 using $NaNO_2$ led to a concomitant spontaneous cyclization to directly give nitrosorbicillactone 21a with the desired relative stereochemistry and the undesired 21b in 4 and 11% yield, respectively. As the overall yield in this cyclization step was low and as the undesired epimer 21b was the major product, this route was not further investigated. As an alternative, we next investigated the possibility of performing the cyclization reaction without a protective group but rather with the required fumaric acid residue already attached (Fig. 3a). Peptide coupling of L-alanine-O*t*Bu (22) with mono-ethyl fumarate 23 in 97% yield. After the removal of the *t*Bu protective group (quant.), esterification of 23 with sorbicillin (1) yielded 25 in 37% yield. The PIFA-mediated oxidation of 25 yielded sorbicillinol derivative 26 in 48%, which was further cyclized to the desired sorbicillactone ethyl ester 27a and the undesired 27b. Unfortunately, the yields and the stereochemical

outcome in this crucial step were unsatisfactory, delivering ~1% of 27a and 2% of 27b. Despite these daunting initial results, we set out to convert 24 into a pfp-activated ester to facilitate the interception of chemoenzymatically prepared 2, in analogy to the successful chemoenzymatic synthesis of model substrate 16a (Fig. 2c). Surprisingly, we did not see any formation of the desired active ester. In a detailed investigation of the reaction course by NMR and HRMS analysis, we proved that an azlactone was formed instead (see supporting information section 2.19, incl. Fig. S1/S2). Since azlactones can act as nucleophiles in Michael-type reactions and as they are generally widely used in organic synthesis[27], we next turned our attention to converting enzymatically produced sorbicillinol (2) with unprotected azlactone 31 to potentially gain direct access to sorbicillactone A (7).

**Chemoenzymatic total synthesis of sorbicillactone A (7)**

The desired azlactone precursor 30 can readily be synthesized by coupling of L-alanine-OMe (28) and mono-ethyl fumarate followed by saponification of both esters in 29 to give 30 in 90% yield over two steps. (Fig. 3b) The ring closure to form azlactone 31 was performed by simply activating the carboxylic acid with EDC. The unstable compound 31 thus prepared was used without further purification and reacted in situ with enzymatically prepared sorbicillinol (2). This directly furnished stereochemically pure sorbicillactone A (7) in 29% yield. The reaction proceeded smoothly and did not show the formation of any undesired stereoisomer. Similar to our observations in the chemoenzymatic synthesis of (+)-epoxysorbicillinol[17], we suppose that the tertiary alcohol function in 2 directs the azlactone to attack from the same side by precomplexation. Regarding the acidity of 31 and the slightly basic reaction conditions, it is reasonable to assume that reaction proceeds via the enolate 31', which coordinates to the C5 alcohol in 2. This leads to chirality transfer from the enzymatically defined C5 stereogenic center in 2 to the C6 stereocenter, while the stereogenic outcome at C9 is determined by the least sterically demanding configuration, resulting in a

**Fig. 3 | Total synthesis of sorbicillactone A (7) and C9 analogs. a** Chemical synthesis of sorbicillactone ethyl esters (**27a/b**). **b** Chemoenzymatic total synthesis of sorbicillactone A (**7**) and structural analogs **33–36** with different C9 substitutions using SorbC (labeled in yellow).

*syn* positioning of the two smallest substituents at C9 (methyl group) and C6 (proton). Encouraged by this stereoselective access to sorbicillactone A (**7**), we evaluated the applicability of this approach to synthesize a set of C9 analogs of **7** (Box Fig. 3b). Therefore, derivatives of fumarylazlactone **31** were synthesized starting from different amino acid methyl esters. Following the same experimental procedures as shown in Figs. 3, 2b, the methyl esters of aminobutyric acid, norvalin, leucine, and isoleucine were coupled to mono-ethyl fumarate. Subsequent deprotection of both esters afforded the analogs of **30** in 82–93% yield. Again, activation with EDC led to the formation of the corresponding azlactones. Their final addition to enzymatically produced sorbicillinol (**2**) afforded the derivatives **33–36** of sorbicillactone A (**7**) in 19–26% yield. Formation of **33–36** showed no changes in the preferred stereochemical outcome compared to the natural product, even with sterically demanding amino acids, such as isoleucine, highlighting the stereochemical resilience of the final reaction step.

When evaluating the overall reaction performance, it is important to note that the crude preparation of azlactone (**31, 33–36**) is directly added to the enzymatic transformation of **1** to **2** in water. The aqueous reaction environment leads to partial hydrolysis of the rather unstable azlactone building block. In addition, **2** can give rise to small amounts of dimeric side-product, such as bisorbicillinol. This, together with the isolation of the target products **7, 33–36** by preparative HPLC, explains the isolated yields. In addition, increasing steric demand at the azlactone unit leads to the observed relative yield reduction of isolated sorbicillactone analogs, ranging from 29% for **7** (methyl substituent) to 19% in **36** (*sec*-butyl substituent).

**Biosynthetic considerations**
Recent work by Watanabe et al. beautifully illuminated the biosynthesis of the fungal natural products fumimycin and lentofuranine[28]. These compounds arise from a phenolic intermediate of the terrein biosynthetic pathway, which, upon oxidative dearomatization, gives a quinone that reacts with fumarylazlactone **31** in a cross-pathway non-enzymatic condensation reaction. Nucleophile **31** is formed by a bimodular non-ribosomal peptide synthetase ALT_2306 (AlSidE). Interestingly, genomes of *Penicillium* sp. containing biosynthetic machinery to produce **1** and **2** discovered by Cox et al.[2,3,5] also harbor highly similar, putative fumarylazlactone NRPSs encoding genes. For example, the genome of *Penicilium rubens* Wisconsin 54–1255 (GenBank GCA_000226395.1, assembly PenChr_Nov2007)[29] encodes the complete set of sorbicillin biosynthetic enzymes, namely the PKS biosynthetic enzymes SorA (accession number B6HNK3.1) and SorB (B6HN77.1), the FAD-dependent monooxygenase SorC (B6HN76.1), as well as the enzyme SorD (B6HNK6.1)[2,3,5], together with an unassigned enzyme with high homology to putative fumarylazlactone NRPS biosynthetic enzymes from *Aspergillus novofumigatus* IBT 16806 (XP_024680046.1), *A. fumigatus* Af293 (XP_748654.2), and *A. udagawae* (XP_043143650.1) (>64% identity at the protein sequence level)[28]. While their interplay to complete the biosynthetic assembly of **7** remains to be experimentally proven, this, together with our chemoenzymatic access to **7** fusing sorbicillinol (**2**) and **31** in a highly stereoselective manner, strongly suggests that **7** is biosynthetically formed analogously. The discovery of highly reactive **31** as a biosynthetic building block by Watanabe et al.[28] thus seems to be frequently utilized by Nature to generate diverse, complex fungal natural product scaffolds.

**Conclusions**
In conclusion, we established an efficient stereoselective total synthesis of **7** and five unnatural C9 analogs (**16a, 33–36**). The route is highly convergent, with only four steps in the longest linear sequence, very short, with excellent

yields (11% starting from 2-methylresorcinol) and stereochemical control. This approach also opens the door for in-depth biological evaluation and structure-activity relationship studies of the sorbicillactones, which are currently ongoing in our laboratory.

## Methods
Instrumentation, materials, and synthetic procedures
See Supplementary Methods in the Supplementary Information.
Characterization of reactive azlactone
See Supplementary Information Section 2.19, incl. Fig. S1/S2.
NMR spectra of synthesized compounds
See Supplementary Data 1

## Data availability
All data generated during the current study are deposited in Supplementary Information and Supplementary Data 1 and are additionally available from the corresponding author on request.

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

## Acknowledgements
We thank Dr. T. Lübken and his team (TU Dresden, Organic Chemistry I) for recording NMR spectra. We thank the DFG for the generous financial support of the work in our laboratory (GU 1233/7-1, INST 269/971-1).

## Author contributions
J.I.M. conducted all experimental work. T.A.M.G. supervised the entire project and secured project funding. Both authors designed the experimental work, analyzed the data, discussed the results, and wrote the manuscript.

## Funding

## Competing interests
The authors declare no competing interest.
