## [Peer Review File · Communications Chemistry]

REVIEWERS' COMMENTS:

Reviewer #1 (Remarks to the Author):

The manuscript by Müller and Gulder describes the first enantio- and diastereoselective synthetic route to sorbicillactone A utilizing an enzyme-triggered approach that was previously developed by the Gulder group and applied for a number of natural product targets. While their previous work was mostly relying on cycloaddition chemistry to trap the enzymatically produced sorbicillinol as key intermediate, thus taking advantage of intrinsic stereospecificity, the herein described target poses a more ambiguous trapping challenge that requires the diastereoselective construction of three contiguous chiral centres as part of a Michael addition.

The manuscript is overall well written and concise, and offers a lot of interesting chemistry surrounding their SorbC biocatalysis module combined with the general dearomatization chemistry en route to sorbicillactone A. Here, both enzyme free strategies (using PIFA as oxidant) and SorbC-catalyzed dearomatizations are discussed with a number of coupling partners, leading to first truncated versions of the natural product (or intermediates), later to the actual secondary metabolite along with a handful of structural analogues (in high enantio- and diastereoselectivity). The story is finalized with a reflection on sorbicillactone A's biosynthesis with references to bioinformatics data that indicate that their azlactone coupling partner may indeed be also the biosynthetic source of the amino acid fragment.

I genuinely enjoyed the report and believe that it is of general interest for a broad readership. However, I felt that the overall story line was a bit convoluted jumping in between PIFA chemistry and the SorbC catalysis. I am not sure if it became more clear if the enzyme-free approaches would be discussed first. Either way, please indicate that PIFA-induced dearomatizations yield 50:50 mixtures of diastereomers and the resulting Michael cyclizations racemic mixtures (or did I get this part wrong?).

One more point of clarification could be added regarding the discussion of azlactone 31. 31 is drawn with an alpha-CH first, then as tautomer with alpha-CH at fumarate terminus. While explained in the supporting information, it would be helpful to introduce both isomers also in the main document (perhaps as 31 and 31'). Please also comment on the acidity of 31. Could an anionic azlactone enolate become relevant under reaction conditions in phosphate buffer (which would probably be a better H-bond acceptor with the dearomatized partner)?

And as a minor point, I totally agree that the presented synthesis is very elegant and streamlined, but the word streamlined (or variations thereof) is perhaps used a little too frequently (i.e. three times on page 2 alone). Apart from these minor corrections, I believe that the manuscript is well suitable for publication in Communications Chemistry, and I would support its acceptance after minor revisions.

Reviewer #2 (Remarks to the Author):

This manuscript describes the development of an efficient, flexible, and stereoselective synthetic approach to sorbicillactones and also sheds light on the biosynthesis of this unusual natural product.

Overall, this research was well designed, executed, and the manuscript is well-written. The supporting information is comprehensive as well. Hence, this referee recommends to accept it for publication after minor revisions.

1) In the developed key transformation of compound 1 to sorbicillactone A (7) and its analogs, the isolated yields for the desired products were moderate, ranging from 19% to 29%. What kind of factors could lead to this modest yield? Such as incomplete consumption of starting material, by-product formation, or product loss during purification. I would suggest the authors include relevant discussions in the manuscript.

Reviewer #3 (Remarks to the Author):

The manuscript by Gulder and co-workers describes a chemo-enzymatic total synthesis of sorbicillactone A, a polyketide natural product with anti-HIV activity. Sorbicillinoids have garnered significant attention from the synthesis community beginning with early biomimetic studies by Nicolaou et. al. in 2000. The Gulder group has contributed to this area previously (2017/2018 ACIEE, 2019 OL, 2022 ACS Catal.), using the monooxygenase SorbC to carry out the key oxidative dimerizations responsible for the synthesis of various members.

In this work, a nitrogen-containing member (sorbicillactone A) is constructed in a very robust and short sequence wherein complete stereocontrol is noted. Key steps of this work include a regio- and stereoselective enzymatic dearomatization of sorbicillin to sorbicillinol and the direct conversion of sorbicillinol into sorbicillactone A by a one-pot addition of various azlactones. Overall, only 4 synthetic steps (LLS) were needed and the overall yield was >10% yield. In addition, the authors prepare some unnatural analogs with different substituents at C9. This work compares quite favorably (and likely exceeds) past fully synthetic routes to this target (2011 OL) in terms of both yield and stereocontrol. The SI is satisfactory and I believe this work would be of interest to the readers of communications chemistry and recommend its acceptance.

To
Communications Chemistry
Editorial Team

Prof. Dr. rer. nat.
Tobias A. M. Gulder

Bearbeiter:
Telefon: 0351 463-34494
Telefax: 0351 463-35506
E-Mail: tobias.gulder@tu-dresden.de

Dresden, January 31st 2024

Dear Reviewers,

We thank you very much for the fast reviewing process and the very positive evaluation of our work. In response to the requests by our reviewers, we have made the following changes / additions to our manuscript:

- Reviewer 1, request 1:

‘I genuinely enjoyed the report and believe that it is of general interest for a broad readership. However, I felt that the overall story line was a bit convoluted jumping in between PIFA chemistry and the SorbC catalysis. I am not sure if it became more clear if the enzyme-free approaches would be discussed first. Either way, please indicate that PIFA-induced dearomatizations yield 50:50 mixtures of diastereomers and the resulting Michael cyclizations racemic mixtures (or did I get this part wrong?).’

→ *Thank you very much for pointing this out! As requested by the reviewer, we have added diastereomeric ratios (d.r. 1:1) to each reactions product of PIFA-mediated oxidations. The stereochemical outcome of the Michael cyclization reactions is presented in boxes in each of the respective Figures (please also see comments at the respective figures in the manuscript file with tracked changes).*

- Reviewer 1, request 2:

‘One more point of clarification could be added regarding the discussion of azlactone 31. 31 is drawn with an alpha-CH first, then as tautomer with alpha-CH at fumarate terminus. While explained in the supporting information, it would be helpful to introduce both isomers also in the main document (perhaps as 31 and 31'). Please also comment on the

acidity of **31**. Could an anionic azlactone enolate become relevant under reaction conditions in phosphate buffer (which would probably be a better H-bond acceptor with the dearomatized partner)?'

→ *We thank the reviewer for these excellent suggestions. Accordingly, we have added all structures of the relevant isomers **31**, **31'** and **31''** to Figure 3 and have adjusted the graphical representation of the Michael addition reaction (now using enolate **31'**, as suggested by the reviewer). We have also added a short statement on acidity to the main manuscript text as follows:*

'Regarding the acidity of **31** and the slightly basic reaction conditions, it is reasonable to assume that reaction proceeds via the enolate **31'**, which coordinates to the C5 alcohol in **2**.'

- Reviewer 1, request 3:

'And as a minor point, I totally agree that the presented synthesis is very elegant and streamlined, but the word streamlined (or variations thereof) is perhaps used a little too frequently (i.e. three times on page 2 alone).'

→ *We absolutely agree with reviewer 1 and adjusted the manuscript accordingly.*

- Reviewer 2, only request:

'In the developed key transformation of compound **1** to sorbicillactone A (**7**) and its analogs, the isolated yields for the desired products were moderate, ranging from 19% to 29%. What kind of factors could lead to this modest yield? Such as incomplete consumption of starting material, by-product formation, or product loss during purification. I would suggest the authors include relevant discussions in the manuscript.'

→ *We thank reviewer 2 for this valuable suggestion and have added the following section to the main manuscript text:*

'When evaluating the overall reaction performance, it is important to note that the crude preparation of azlactone (**31**, **33-36**) is directly added to the enzymatic transformation of **1** to **2** in water. The aqueous reaction environment leads to partial hydrolysis of the rather unstable azlactone building block. In addition, **2** can give rise to small amounts of dimeric side-product, such as bisorbicillinol. This, together with the isolation of the target products **7**, **33-36** by preparative HPLC, explains the isolated yields. In addition, increasing steric demand at the azlactone unit leads to the observed relative yield reduction of isolated sorbicillactone analogs, ranging from 29% for **7** (methyl substituent) to 19% in **36** (*sec*-butyl substituent).'

We hope we have sufficiently addressed all concerns raised by the reviewers. In case you need any additional information, please let us know.

We are looking forward to your final decision on our manuscript.